# Arsenate Adsorption on Fly Ash, Chitosan and Their Composites and Its Relations with Surface, Charge and Pore Properties of the Sorbents

**DOI:** 10.3390/ma13235381

**Published:** 2020-11-26

**Authors:** Agnieszka Adamczuk, Weronika Sofinska-Chmiel, Grzegorz Jozefaciuk

**Affiliations:** 1Institute of Agrophysics PAS, Doswiadczalna 4 Str., 20-290 Lublin, Poland; a.adamczuk@ipan.lublin.pl; 2Analytical Laboratory, Institute of Chemical Sciences, Faculty of Chemistry, Maria Curie Sklodowska University, Maria Curie Sklodowska Sq. 3, 20-031 Lublin, Poland; wschmiel@poczta.umcs.lublin.pl

**Keywords:** fractal, mercury intrusion, isotherm, equilibrium, kinetics, thermodynamics, waste management

## Abstract

One of the ways to recycle millions of tons of fly ash and chitin wastes produced yearly is their utilization as low-cost sorbents, mainly for heavy metal cations and organic substances. To improve their sorption efficiency, fly ashes have been thermally activated or modified by chitosan. We aimed to deeply characterize the physicochemical properties of such sorbents to reveal the usefulness of modification procedures and their effect on As(V) adsorption. Using low temperature nitrogen adsorption, scanning electron microscopy, mercury intrusion porosimetry, potentiometric titration and adsorption isotherms of As(V) anions, surface, pore, charge and anion adsorption parameters of fly ash activated at various temperatures, chitosan, and fly ash modified by chitosan were determined. Arsenate adsorption equilibrium (Langmuir model), kinetics (pseudo-second order model) and thermodynamics on the obtained materials were studied. Neither temperature activation nor chitosan modifications of fly ash were necessary and profitable for improving physicochemical properties and As(V) adsorption efficiency of fly ash. Practically, the physicochemical parameters of the sorbents were not related to their anion adsorption parameters.

## 1. Introduction

Among the basic problems of today’s civilization is the utilization of municipal, industrial, and agricultural wastes. From an ecological point of view, the best solution is to reuse this type of waste as alternative energy sources, concrete or asphalt additives, soil fertilizers, conditioners etc. [1,2]. A recently developing direction is the reuse of waste products for sorption of dyes, heavy metals, and other environmental pollutants [3,4,5]. Novel low-cost adsorbents include, among others, biosorbents originated from plant and food production and adsorbents derived from industrial wastes [6,7].

European Coal Combustion Products Association (ECOPA), based on monitoring 15 European countries, announced that the production of fly ash in 2016 reached over 25 MT [1]; therefore, its management is of interest to societies and scientists. Recently, fly ash has been applied in the production of sorbents for water, sewage, soil, and exhaust fumes treatment to remove heavy metals [2,6,7], phosphates [8,9], humic acid [10,11], pesticides [12,13,14], dyes [15,16,17], fluoride [18], boron [19], ammonium [20], tannic acid [21], petroleum compounds [22,23], radioactive pollutants [23], chemical oxygen and suspended solids [24,25]. A frequent procedure applied to fly ash, aiming at improving its sorption properties, is a heat treatment called thermal activation [26,27].

Another common waste product is chitin, from which chitosan is produced through low-cost chemical treatment [28]. The molecular weight of chitosan, decreasing with the degree of deacetylation, ranges from 3800 to 500,000 g/mol [29]. Chitosan and its composites have been used for the removal of inorganic and organic pollutants from water [30,31,32,33,34]. Xie et al. [35] showed that synthetic zeolite from a coal fly ash modified by chitosan adsorbed more phosphate than the zeolite itself. Ocinski and Mazur [36] applied Fe-Mn waste oxides coming from water deironing entrapped into a chitosan matrix for arsenic sorption. Chitosan modified fly ash was used as chromium sorbents from water [37].

Many surface and structural properties have been used to explain the adsorption behavior of different materials [38,39,40,41,42]. Since most of the related literature concerns heavy metal cations and organic adsorbates, we concentrate on anion adsorption. We selected arsenic(V) anions because of the harmful effect of arsenic on living organisms. The International Agency for Research on Cancer classifies arsenic in the first group of carcinogenic metals [43], which results from the competitive activity of As(V) towards P(V) anions [44]. Various sorbents have been applied to remove arsenic from waters: active carbon [45,46,47,48,49], manganese, iron, and aluminum oxides [50,51], titania and titanium-based metal and metal oxide nanocomposites [52], and magnetite modified fly ash [53].

The present paper aims to analyze the efficiency of As(V) adsorption on thermally activated fly ash coated by chitosan to revive a usefulness of these waste products in environmental protection. The next aim is to determine the physicochemical properties of the adsorbents, find their changes under modification procedures and to relate them to As(V) adsorption behavior.

## 2. Materials and Methods

Chitosan (abbreviated hereafter as CS) of average molecular weight 358.2 kDa and degree of deacetylation of around 70% was purchased as a powder from Sigma-Aldrich, (Merck KGaA, Darmstadt, Germany). Fly ash (FA) belonging to class F according to the American Society for Testing Materials [54] containing 9.7% SiO_2_, 27.5% Al_2_O_3_, 5.93% Fe_2_O_3_ and 11.0% CaO was obtained from Kozienice Power Plant (Poland). The As(V) solution was prepared by dissolving Na_2_HAsO_4_ (analytical grade, POCH Poland) in deionized water. Since solution pH has a crucial effect on ion speciation, influencing the efficiency of sorption, the As(V) solution was adjusted with hydrochloric acid to a pH of 6.0, at which H_2_AsO^4−^ anion dominates [55].

### 2.1. Preparation of the Adsorbents

Temperature activated fly ashes were prepared by 1 h combustion of an initial fly ash (FA) in a muffle furnace at 773, 973 and 1173 K (denoted further as FA500, FA700, FA900, respectively). The materials were then adjusted to pH = 6 with hydrochloric acid and washed with distilled water until a negative reaction with AgNO_3_.

Parts of the initial and temperature activated fly ash were coated with chitosan. To do this, 20 g of chitosan was jellified in a solution containing 1500 cm^3^ of water and 120 cm^3^ of 99.5% acetic acid by 24 h mechanical stirring (1000 rpm) at room temperature. Next, 80 g of fly ash was added to the (still agitated) chitosan gel and the pH of the resulting mixture was slowly adjusted to 9 with 5% NaOH solution. When the requested pH did not change within 10 min, the suspension was agitated in the next 30 min, filtered, and washed with distilled water until the pH of the supernatant was neutral. The obtained material was dried for 48 h at 323 K, gently ground in a mortar and 1 mm sieved. The obtained sorbents are named as FACS; FA500CS; FA700CS and FA900CS, respectively. The treatments and the adsorbents obtained are shown in Scheme 1.

### 2.2. Scanning Electron Microscopy

Thousand times magnified images of the studied adsorbents were taken in ten replicates using Phenom ProX desktop SEM (Thermo Fisher Scientific, Waltham, MA, USA). Representative images were selected.

### 2.3. Surface and Pore Properties of the Adsorbents

#### 2.3.1. Nitrogen Adsorption

Low temperature nitrogen adsorption/desorption isotherms relating the amount of adsorption, A (kg kg^−1^), against the relative nitrogen pressure, p/p_0_, were measured for the studied adsorbents using ASAP 2405 sorptomate (Micromeritics Inc., Gwinnett County, GA, USA) in three replicates. The maximum deviations between the replicated isotherms were 2.7%.

Surface area values were calculated from adsorption data using the linear form of the Aranovich isotherm [56]:p/p_0_/[A(1 − p/p_0_)^1/2^] = 1/(A_m_ C) + p/p_0_/A_m_,(1)
where A_m_ (kg kg^−1^) is the statistical monolayer capacity and C is a constant.

In contrast to the standard Brunauer–Emmett–Teller (BET) model [57], the Aranovich isotherm is thermodynamically correct. It allows for the presence of vacancies in the adsorbed layer and therefore the surface areas derived from the Aranovich model are larger than those from BET. The Aranovich equation fits the experimental polymolecular adsorption data over a broader range of relative pressures (~0.05 < p/p_0_ < 0.8) than the BET does (~0.05 < p/p_0_ < 0.35). After calculating A_m_ values from the slopes of the linearity range of Equation (1), the surface area, S (m^2^ kg^−1^), of the adsorbents was calculated as:S = Nω A_m_/M,(2)
where: N (mol^−1^) is the Avogadro number, M (kg) is molecular weight of nitrogen and ω (m^2^) is the area occupied by a single nitrogen molecule, assumed to be 1.63 × 10^−19^ m^2^.

The surface fractal dimension, D, was calculated from adsorption isotherm, from the slopes of the linear parts (if any) of the ln-ln plots of adsorption (A) vs. adsorption potential, using the equation [58]:ln(A) = C − (1/m)ln(RTln(p_0_/p)),(3)
where: C is a constant, RTln(p_0_/p) is the adsorption potential, R (J mol^−1^ K^−1^) is universal gas constant, T (K) is the temperature of the measurements, and the parameter m is related to the surface fractal dimension of the sample.

The magnitude of the parameter 1/m distinguishes two possible adsorption regimes: when 1/m < 1/3, the adsorption occurs within the van der Waals regime and the surface fractal dimension is D =3(1 − 1/m). Alternatively, for 1/m > 1/3, the adsorption is governed by the capillary condensation mechanism and D = 3 − 1/m.

The total nanopore volume, v_t_ (m^3^), was taken from the desorption branch as the amount of liquid nitrogen accumulated in the adsorbent in the p/p_0_ range between 0.40 and 0.98. Selection of the lower limit meets a frequent assumption that below p/p_0_ around 0.35 surface adsorption processes dominate and the condensation of vapors in pores occurs at higher relative pressures. The upper limit was p/p_0_ common for all samples. The lower p/p_0_ limit corresponds to 1 nm and the upper one to 500 nm pore radius r (m), calculated from the Kelvin equation [59]:r = 2M σ cosα/ ρRT ln(p_0_/p),(4)
where: M is the molecular mass of nitrogen, σ is its surface tension, ρ its density, α is a liquid nitrogen–solid contact angle (assumed here to be zero).

The average pore radius, r_av_ (m), was estimated from volumetric fractions of nanopores, f(r). The pore volume at a given (radius-corresponding) pressure was treated as a sum of pore volumes, v_i_(r_i_), of the radii r_i_ ≤ r:v(r) = ∑_i=1_^n^ v_i_(r_i_).(5)

Dividing the above equation by the total pore volume one has:v(r)/v_t_ = ∑_i=1_^n^ v_i_(r_i_)/v_t_ =∑_i=1_^n^ f(r_i_),(6)
and the average pore radius, r_av_ (m), is:r_av_ = ∑_i=1_^n^ r_i_ f(r_i_).(7)

#### 2.3.2. Mercury Intrusion Porosimetry

Mercury intrusion porosimetry (MIP) tests were performed for pressures ranging from c.a. 0.1 to 200 MPa (pore radii from c.a. 10.0 to 3.8 × 10^−3^ µm) using an Autopore IV 9500 (Micromeritics, Gwinnett County, GA, USA) porosimeter for three replicates of the studied adsorbents (maximum deviations were up to 4.8%). To avoid measuring extremely high pore volumes for powder forms (connected with different side effects) all adsorbents were studied as “aggregates” [60]. To do this, the adsorbents were fully saturated with water, placed in glass (nonporous) tubes and dried. Since most of the aggregates were very fragile, they were not withdrawn from their own tubes for mercury intrusion testing. To better characterize chitosan properties, in MIP tests, CSF (flake), not powdered chitosan, was studied additionally. To obtain CSF, the initial chitosan material was dissolved in acetic acid, adjusted with NaOH to pH = 9, washed with distilled water and then dried but not powdered (similar procedure as for CS modification of FA). The intrusion volumes were measured at stepwise increasing pressures allowing equilibration at each pressure step. The volume of mercury V (m^3^ kg^−1^) intruded at a given pressure P (Pa) gave the pore volume that can be accessed. The intrusion pressure was translated to an equivalent mesopore radius R (m) following the Washburn equation [61]:P = −B σ_m_ cosα_m_/R,(8)
where: σ_m_ is the mercury surface tension, α_m_ is the mercury/solid contact angle (taken as 141.3° for all studied materials) and B is a shape factor (equal to 2 for the assumed capillary pores).

The average mesopore radius was calculated from a normalized pore size distribution, χ(R), estimated from the dependence of V vs. R, and expressed in the logarithmic scale [62]:χ(R) = 1/V_max_ dV/dlog(R).(9)

Knowing χ(R), the average pore radius, R_av_, was calculated as:R_av_ = ∫R χ(R) dR.(10)

The pore surface fractal dimension, D_s_, was determined from the slope of the linear part (if any) of the dependence of log(dV/dR) on logR [63]:Ds = 2 − slope.(11)

The penetration thresholds, PT, (i.e., the points at which mercury starts to enter the internal body of the adsorbents) were approximated by the pore radii at which the second derivative of pore volume vs. log radius equals zero [60]:d^2^V/d(logR)^2^ = 0.(12)

The intra-aggregate pore volume, V_ia_, was taken as the volume of mercury that intruded into the pores of lower radii than PT (at higher pressures than those corresponding to the PT).

The intra-aggregate pore radius, R_ia_, was calculated similarly to the average mesopore radius (Equations (9) and (10)) from a part of the MIP curve below PT.

#### 2.3.3. Potentiometric Titration

Potentiometric titration curves for the suspensions of the studied adsorbents were taken using auto-titrator SM Titrino 702 (Metrohm, AG, Herisau, Switzerland). Prior to the titration, the studied adsorbents were adjusted with H_2_SO_4_ to pH values of 3 and 5, filtered, washed with distilled water and air dried. The sulfuric acid was used because it did not jellify chitosan. Suspensions of 0.1 g of the chitosan and 1.0 g of the other adsorbents in 20 cm^3^ 0.01 M Na_2_SO_4_ were slowly titrated (1 µL increments per 1 min) under nitrogen atmosphere to pH = 9 with 0.1 mol·dm^−3^ NaOH dissolved in 0.01 M Na_2_SO_4_. Parallelly, 20 cm^3^ of 0.01 mol·dm^−3^ Na_2_SO_4_ (supernatant) was titrated as well. The pH was registered in steps of 0.1 unit. Titration curves were registered and expressed as dependencies of the added base on pH. All adsorbents were titrated in three replicates with deviations not exceeding 5.6%. Since we planned to perform As(V) adsorption measurements at 293, 313 and 333 K, we titrated all adsorbents at the same temperatures as well. As the titration curves measured at various temperatures did not practically differ, we did not replicate the titrations at 313 and 333 K.

Surface charge vs. pH dependencies were determined from titration data. As the amount of the base consumed by the whole suspension at any pH is used for neutralization of acidic surface functional groups of the solid and of the acids present in the supernatant, the titration curve of the suspension, minus the titration curve of 0.01 M Na_2_SO_4_, was taken as the titration curve of the solid phase. The latter curve is the dependence of surface charge increase on pH [64]. It can be used to find the dependence of the net surface charge on pH by shifting along the charge axis to meet any point at which the total surface charge is known. We shifted the experimental solid titration curves to zero surface charge assuming that it occurs at isoelectric points of the studied materials i.e., at pH = 7.1 for chitosan and at 8.2 for the original and the temperature activated fly ashes [36]. For chitosan modified fly ashes, the point of zero charge (pH 7.7) was arbitrarily assumed to be located between its values for CS and FA’s.

We applied rather high ionic strength (0.01 M Na_2_SO_4_) during the titration. At high ionic strengths, the concentration of any ion near the charged surface approaches its concentration in the bulk solution due to the shrinking of diffuse double layer. In this case, the surface concentration of protons is closer to their concentration in the solution and the surface charge is better approximated by the amount of the titrant consumed. High ionic strength also allows for a better development of variable surface charge because at high ionic strengths dissociation of surface groups is less hindered by electrostatic effects. Applied high concentrations of the titrant (0.1 mol·dm^−3^ NaOH) suppressed a dilution of the suspension during titration that minimizes changes in the suspension effect (depending on the suspension concentration), which could affect the measured pH. The above aspects are discussed in ref. [65].

### 2.4. As(V) Adsorption Tests

#### 2.4.1. Adsorption Equilibrium

Suspensions containing 0.20 g of the studied adsorbents and 20 cm^3^ portions of As(V) solutions of pH = 6 and concentrations ranging from 50 to 1800 mg dm^−3^ were shaken (180 rpm, 5 h) at 3 different temperatures (293, 313 and 333 K). In the filtered solutions, pH and As(V) were determined. The concentration of As(V) was analyzed spectrophotometrically at 870 nm with (NH_4_)_6_Mo_7_O_24_·H_2_O as the complexing agent using the Cary 60 spectrophotometer (Agilent Technologies, Santa Clara, CA, USA). The measurements of adsorption equilibriums were replicated thrice. The maximum deviations between the measured concentrations were smaller than 2.3%. The pH was measured using Elmetron (Poland) pH meter. The preliminary leachability tests, performed to check eventual presence (desorption) of As(V) in fly ashes, detected no As(V) in the leachates.

Amount of the adsorbed As(V) by the unit mass of the adsorbent, a (kg kg^−1^), was calculated as:a = V(c_0_ − c_e_)/m,(13)
where: c_0_ is the initial concentration of As(V) in the aqueous phase (kg·m^−3^); c_e_ is its equilibrium concentration (kg m^−3^); V is the volume of the solution (m^3^); m is the mass of the adsorbent (kg).

We found that the experimental adsorption data were best fitted to the Langmuir [66] isotherm; therefore, from its linear form:c_e_/a = 1/(a_m_K_L_) + c_e_/a_m_(14)
the adsorption parameters: adsorption capacity, a_m_ (kg kg^−1^), and Langmuir adsorption constant K_L_ (m^3^ kg^−1^) were calculated.

#### 2.4.2. Adsorption Thermodynamics

To evaluate the adsorption thermodynamics of dyes onto the adsorbent, Equation (5) is widely used [67]:ΔG° = ΔH° + TΔS°,(15)
where: ΔH° is the free enthalpy and ΔS° is the entropy of the adsorption process and the free energy of the system, ΔG° (J mol^−1^) is defined by:ΔG° = −RTlnK_0_,(16)
where: K_0_ is taken as the adsorption distribution coefficient K_D_ = a/c_e_ determined for each sorbent at the smallest As(V) concentration.

Combining Equations (15) and (16):−RlnK_D_ = ΔH°/T + ΔS°,(17)
and having a set of K_D_ values estimated at different adsorption temperatures, the ΔH° values can be estimated from the slope of the linear fit of the dependence of lnK_D_ against 1/T.

#### 2.4.3. Adsorption Kinetics

Suspensions containing 0.20 g of the adsorbents in 20 cm^3^ portions of As(V) solutions of pH = 6 and concentrations 100; 200 and 300 mg dm^−3^ were shaken (180 rpm, 293 K) at different time intervals (1; 3; 5; 10; 15; 20; 30; 60; 120 and 180 min) and filtered. In the filtrates, As(V) concentration and pH were measured similarly as in adsorption tests. The measurements of adsorption kinetics were replicated thrice. The maximum deviations between the measured concentrations were smaller than 3.5%.

As(V) content in the adsorbent after time t, q_t_ (kg kg^−1^), was calculated as:q_t_ = V(c_0_ − c_t_)/m,(18)
where: c_0_ (kg m^−3^) is the initial As(V) concentration in the solution and c_t_ (kg m^−3^) is its concentration after time t (s).

Since the experimental data were best fitted to the pseudo-second order kinetic model (PSO) therefore from its linear form:t/q_t_ = t/q_e_ + 1/(kq_e_^2^)(19)
the kinetic parameters: the equilibrium adsorption for a given c_0_, q_e_ (mol kg^−1^ s^−1^) and equilibrium constant for a given c_0_, k (mol kg^−1^ s^−1^) were calculated.

### 2.5. Analysis of Models Validity

Except for the linear regression coefficients of the experimental equilibrium adsorption and kinetic data fits to linear forms of the Langmuir and PSO models, respectively, to analyze the validity of the model approximations, the relative root mean square error (RRMSE) was used:RRMSE = 1/n [∑_i=1_^n^ (y_i_ − y_i,model_)^2^/(y_i_)^2^]^1/2^(20)
where y_i_ is the experimental value, y_i,model_ is its corresponding value calculated from a model and n is the number of sampling of the y_i_(y_i,model_) relationship.

## 3. Results

### 3.1. Solid Phase Characteristics from SEM Images

Selected SEM images, which in our opinion are the best to illustrate the processes of temperature activation and chitosan modification of fly ashes, are presented in Figure 1.

In the FA image, spherical particles dominate, which may belong to two groups: cenospheres (hollow ceramic particles made largely of alumina and silica) and/or plerospheres (hollow particles of large diameter filled with smaller size ones) produced at temperatures of 1723 to 2023 K during coal burning in thermal power plants. In the temperature treated fly ash (FA900), the spheres are larger and more irregular. In the FA material, spongy structures (surrounding by black circles) possibly belonging to unburned carbon are observed, which disappear after heating. In the FACS images, irregular chitosan particles are seen, upon which (and probably inside) the ceramic spheres are glued, which is better seen in the magnified FACS image.

### 3.2. Surface and Pore Properties of the Adsorbents

#### 3.2.1. Surface and Nanopore Properties Derived from Nitrogen Adsorption Isotherms

Low temperature nitrogen adsorption isotherms of the studied adsorbents are presented in Figure 2. To emphasize details, no desorption branches are shown. This figure is divided into two parts: left part shows low p/p_0_ range and right part shows high p/p_0_ range.

In general, the nitrogen adsorption for all adsorbents is very low. The highest adsorption occurs for FA and it decreases consecutively with increasing fly ash activation temperature. At low pressures, adsorption isotherm of chitosan is located between FA700 and FA900; however, at high pressures, it sharply increases, approaching the adsorption values for FA at the end. Chitosan modification diminishes nitrogen adsorption on all fly ashes but FA900.

Fractal plots for the studied sorbents are illustrated in Figure 3 (the points only belonging to the linear fractality ranges are included).

The slope of the linear fits is the highest for CS and the lowest for FA adsorbent. Broad fractality ranges for fly ashes shorten after chitosan modification. Table 1 summarizes numerical values of surface parameters of the studied adsorbents derived from the isotherms.

The specific surface areas of all adsorbents are very low. The largest surface area occurs for FA and it decreases consecutively with increasing fly ash activation temperature. Similarly, the increase in activation temperature leads to decrease in fly ash nanopore volume. However, the increase in activation temperature enlarges the average pore radius of fly ash. The fractal dimensions of fly ash vary between 3 (rough and complicated surface) and 2 (flat, two-dimensional surface). The roughest surface belongs to FA. Increase in activation temperature consecutively smoothens the fly ash surface, which is consistent with increasing nanopore radii. Chitosan modification diminishes surface areas for all fly ashes but FA900. The latter sorbent has a lower surface area than chitosan itself, which can explain the above effect. The volume of 10–500 nm nanopores and average nanopore radius of chitosan are higher than for fly ash and so the chitosan modification increases pore volumes and nanopore radii of all fly ashes. Since the smoothest surface belongs to the chitosan, chitosan modification consecutively smoothens the surface of fly ash–chitosan composites.

#### 3.2.2. Mesopore Properties of the Adsorbents Derived from Mercury Intrusion Tests

Mercury intrusion porosimetry curves along with mesopore size distribution functions for the studied sorbents are shown in Figure 4.

Mesopore surface fractal plots for the studied sorbents are illustrated in Figure 5 (only those points belonging to the linear fractality ranges are included).

The linearity ranges detected for all adsorbents are in the range of very fine pores. Numerical values of mesopore parameters of the studied adsorbents derived from mercury intrusion curves are summarized in Table 2.

The highest mesopore volume V and average radius R_av_ were found for chitosan in the form of powder (CS) and the lowest for chitosan flakes CSF). The mesopore volume decreases consecutively with increasing fly ash activation temperature. Modification of fly ash with chitosan increases the pore volume and markedly increases the average mesopore radius. As seen from mesopore size distribution functions, most of the mesopores locate in the large pore range, reflecting mostly a mutual organization of the largest adsorbent particles into an aggregate network. The penetration threshold of fly ash markedly increased after chitosan modification. Chitosan modified fly ash has higher intra-aggregate pore volumes V_ia_ and markedly higher average radii R_ia_ than not modified ones. The pore surface fractal dimensions calculated for all adsorbents but FA900 are higher than three, which is theoretically restricted. Such high “fractal dimensions” may result from the specific structure of the pores: if the large pores are accessible through narrower entrances, the large pore volume is attributed to the radius of the entrance and because all volume is treated as belonging to a long capillary in a cylindrical pore model, the dV/dR is also higher and gives D_s_ values higher than three [68]. No evident trends were observed in changes in “mesopore surface fractal dimension” due to the adsorbents’ treatment. The lowest fractal dimension of FA900 indicates that this adsorbent has the flattest and the least complicated mesopore surface.

The chitosan powder (CS) aggregate has the highest values of all mesopore parameters, but its intra-aggregate volume is the lowest. The chitosan flakes (CSF), however, have extremely small mesopore volume and radii. As this can be observed from MIP curves (Figure 4a) and from parallel fractal plots (Figure 5), the finest pores of radii smaller than 10^−2^ µm are almost identical for both CS and CSF samples. We assumed that these pores characterize the structure inside the chitosan body. A similar approach was used for the other adsorbents. The finest pores’ (material pores, MP) parameters calculated for all adsorbents are presented in Table 3.

Temperature activation of fly ash decreases the volume of pores present inside the material. The FA900 sample has the smallest material pore volume and the highest average pore radius. No evident trends in changes in material pore volume and radii after chitosan modification was observed. Note that both CS and CSF specimens have almost identical parameters of material pores.

#### 3.2.3. Surface Charge of the Adsorbents Derived from Potentiometric Titration

Dependencies of surface charge on pH, obtained from titration curves of the studied adsorbents pretreated at two different pH values, are shown in Figure 6. Since the surface charge was practically not affected by the temperature of titration, this figure contains average curves for particular groups of the adsorbents measured at all titration temperatures.

Values of surface charge at pH 6, Q_pH6_, (the same pH as used for As(V) adsorption measurements) read from pH/charge curves (Figure 6) are presented in Table 4. This table also presents the surface charge densities (SCD) calculated from surface charge and surface area values.

The amount of surface charge measured for pH = 3 pretreated adsorbents significantly differs from that measured for the adsorbents pretreated at pH = 5. The surface charge is higher for pH = 5 than for pH = 3 pretreated fly ash. The surface charge measured for pH = 3 pretreated chitosan is extremely high (the reason that only a part of the respective curve is depicted in Figure 6) and much less surface charge developed after its pretreatment at pH = 5. The surface charge appears to decrease slightly with increasing fly ash activation temperature. Of course, due to the high charge of the chitosan, chitosan modified fly ashes exhibit much higher charge than the fly ashes themselves. The surface charge appears to decrease slightly with increasing fly ash activation temperature. The surface charge density of 1 Cm^−2^ is equivalent to six elementary charges per one square nanometer. The much higher values presented in Table 4 seem unrealistic.

### 3.3. Adsorption of As(V) at Different Temperatures

#### 3.3.1. Adsorption Equilibrium

Average As(V) adsorption isotherms on the studied materials at different temperatures are illustrated in Figure 7. The upper part of this figure shows the isotherms for fly ashes and the lower part contains isotherms for all chitosan-containing adsorbents. To make the picture clearer, the isotherms for particular adsorbents are separated and shifted against the c_eq_ axis. To construct the average isotherm, the adsorption, a, was read at the same c_eq_ from each of the experimental isotherm replicate and then averaged. These average isotherms were constructed only for illustration purposes and not for data elaboration, for which raw data were used.

Generally, the As(V) adsorption is higher on fly ash than on chitosan and on chitosan modified fly ash and the isotherms of the latter two groups seem to be steeper at low adsorbate concentrations and flatter at higher concentrations. Adsorption parameters obtained from the Langmuir equation fit are presented in Table 5.

The adsorption capacities for the studied adsorbents follow the trends of adsorption described above. The relative root mean square error of the approximation of the adsorption data by the Langmuir equation is presented in Figure 8.

Despite very good correlation coefficients for the Langmuir equation fit (see Table 5), the approximation of the experimental data by the Langmuir model may involve significant errors, especially at low adsorbate concentrations.

The significant increase in adsorption constants with increasing adsorption temperature suggests an endothermic character of the process that is expressed by positive values of free enthalpies calculated from Equation (17) and shown in Table 6.

#### 3.3.2. Adsorption Kinetics

The kinetics curves of As(V) adsorption on the studied materials at 20 °C for different initial As(V) concentrations are illustrated in Figure 9 (the upper part shows the curves for fly ashes and the lower part contains the curves for all chitosan-containing adsorbents). To make the picture clearer, the curves for particular adsorbents are separated and shifted against the c_t_ axis. Again, as this was done for adsorption data, to construct the average kinetic curve, the q_t_ was read at the same time, t, from each of the experimental replicate and then averaged. These average curves were constructed only for illustration purposes and not for data elaboration, for which raw data were used.

Kinetic parameters calculated from the above curves fitted to the pseudo-second order linear equation are presented in Table 7.

An increase in the equilibrium adsorption with an increase of the initial As(V) concentration is obvious. Any evident general dependence of the kinetic constant and *c*_0_ was observed for all sorbents together. The relative mean root square error of the approximation of the kinetic data by the pseudo-second order equation is presented in Figure 10.

Despite very good correlation coefficients for the PSO equation fit (Table 7), the approximation of the experimental data by the PSO model can bring significant errors, especially at low equilibrium times.

## 4. Discussion

### 4.1. Physicochemical Parameters of the Sorbents

A general decrease in the surface area and nanopore volumes of fly ash under increasing thermal treatment temperature was observed in nitrogen adsorption studies (Table 1). Broadening of fine pores of fly ash along with the decrease in surface fractal dimension (flattening of the surface) are consistent with the above changes. These observations were certified by the decrease in material pore volume observed in mercury intrusion measurements (Table 3) that may be due to closing the finest pores accompanying the material melting at higher temperatures. Analysis of the SEM images helps to explain the above changes. The temperature treatment transforms some of cenospheres into “potatoes”, probably through the melting and collapse of the molten ceramic material and removes spongy structures of unburned carbon that should have a large surface area and porosity possibly due to their oxidation in the oxygen-containing atmosphere of the process. Chitosan has a lower surface area than fly ashes; therefore, their modification by chitosan decreases the surface area of the composites. As seen from SEM images, chitosan may cover parts of fly ash particles, making their surfaces less available. We think that during the modification procedure a part of fly ash surface is covered by chitosan gel and/or some fly ash material is immersed within it. The drying of the above mixtures fastens adhesion between fly ash and chitosan and such agglomerates endure immersion in water. A fraction of the adsorbing surface of both fly ash and chitosan is blocked, and the adsorption is smaller compared to a situation when both adsorbents could react independently.

Contrary to the finest pores present inside the materials (MP), the intra-aggregate pore properties measured by mercury intrusion (Table 2) characterize the bulk of the material bed. Higher mesopore volume, average radius, and penetration threshold after chitosan modification of fly ash suggest higher permeability of fly ash–chitosan composites beds. Such materials can be more useful in column processes where better flow of fluids is very important. The mesopore properties, however, are of little importance in our batch experiments of As adsorption.

The surface charge measured by potentiometric titration being higher for pH = 5 than for pH = 3 pretreated fly ash (Table 4) is most probably due to more intensive removal of the finest (amorphous) material at lower pH. Such compounds exhibit the highest surface charge. Contrary, the chitosan surface charge was much higher for pH = 3 than for the pH = 5 pretreated sample. Since the swelling of chitosan hydrogels increases at decreasing pH values due to the electrostatic repulsion between the protonated amino groups [69], more internal spaces are available for protons at lower pH. Titration beginning at pH = 3 is performed for larger proton adsorbing surfaces and the titration beginning from pH = 5 holds for much smaller surfaces. If the surface would not expand by swelling, both titration curves should coincide at a higher pH. The above observations may be of importance for the methodology of charge measurements by potentiometric titration in which sample solubilization and swelling may significantly alter the results and their interpretation.

### 4.2. Anion Adsorption Characteristics

An increase in adsorption capacities with temperature of the adsorption process (Table 5) may be explained by the decrease in the thickness of the solvent layer surrounding the adsorbent surface, which increases ions diffusion and/or an increase in the number of active centers as a result of the disruption of some internal bonds near existing sites [70,71,72,73]. Contrary to As(V) adsorption capacities, adsorption constants appear to be the lowest for fly ashes, higher for chitosan and the highest for FACS adsorbents. The adsorption constants increase with the increase in the temperature of the reaction indicating that the adsorption process is faster at higher temperatures. A similar effect was obtained by Zubair [74] for Cr(III) adsorption on waste biomass.

We recall here that free enthalpies shown in Table 6 were calculated placing K_0_ as the adsorption distribution coefficient, K_D_ = a/c_e_, determined from a single point of adsorption isotherm at the lowest As(V) concentration. However, this frequent approach tacitly assumes that a/c_e_ is the same for the other points at lower adsorbate concentrations, i.e., that the adsorption obeys Henry’s law. In most cases it may be assumed only when c_e_ approaches zero, which means that a/c_e_ should be measured at very low equilibrium concentrations that are hardly realized in practice. As it was suggested by Khan and Sing [75], more reliable K_D_ values can be estimated by the extrapolation of ln(a/c_e_) vs. c_e_ dependence to c_e_ = 0, as depicted for adsorption data for selected adsorbents studied in this paper, shown in Figure 11. The intercepts of the linear fits plotted for three points of the above dependencies located at the lowest concentrations (drawn with dashed lines) will be used further for calculation of new distribution coefficients.

In the above figure, data for the FA500 333K sample also illustrate the effect of measuring errors on estimation of *K_D_* from Khan and Sing plots (Khan and Sing curve should monotonically increase with decrease in c_e_, tending to reach a plateau at very low concentrations). It is evident that the intercept of the above linear fit depends on the location of the experimental points, i.e., on concentrations at which the adsorption was measured.

From the intercepts of the linear fits (dashed lines) plotted for three points of the above dependencies located at the lowest concentrations, we estimated new *K_D_* values, *K_D_*_(K&S)_, for all studied adsorbents.

For the zero-solution concentration, the Langmuir equation can be written as:a = a_m_ K_L_ c_e_,(21)

Since the K_D_ value is equal to a/c_e_ at c_e_ = 0, the above equation defines the distribution coefficient (named here K_D(L)_) as:K_D(L)_ = a/c_e_ = a_m_ K_L_.(22)

We think that the most reliable values of distribution coefficients can be derived by Equation (22) using adsorption capacity and the Langmuir adsorption constant, both of which can be estimated with rather high accuracy even if adsorption data at very low concentrations are lacking. In our experimental system, the Langmuir equation was best fitted to the experimental data, and thus we had no doubts that the use of Equation (22) for estimation of the distribution constants was the best choice. Of course, the theoretical Langmuir isotherms plotted in Khan and Sing coordinates for very low solution concentrations have intercepts equal to log(a_m_ K_L_). It is necessary to underline that the Freundlich isotherm (defined as a_m_K_F_ c_e_^1/k^), the second most popular one in the description of adsorption from solutions, is useless for the estimation of K_D_: a/c_e_ tends to infinity at c_e_ approaching zero. Since most of adsorption data (including Freundlich-like), at least at low and moderate concentrations, can be rather well fitted to the Langmuir model, we think that the estimation of K_D_ from the Langmuir fit is a good choice—it seems to be least sensitive towards experimental uncertainties.

In our opinion, the biggest weakness of using the distribution coefficient for the calculation of thermodynamic functions is that one can calculate different K_D_ for principally identical adsorption systems, for example for the same adsorbent diluted with completely inactive (nonadsorbing) material. This problem can be solved by expressing the activity of the adsorbed species, a_surf_, not as its amount per unit mass of the adsorbent but as the fraction of the adsorbent occupied by the adsorbate, a_surf_ = a/a_m_ as it is achieved in the theory of heterogeneous reactions. Now the thermodynamic equilibrium constant K_0_ defined as:K_0_ = a_surf_/a_sol_(23)
where a_surf_ is the activity of the adsorbate in the solid phase and a_sol_ is its activity in the solution at equilibrium, for low solution concentrations (where a_sol_ can be replaced by c_e_) may be written as:K_0_ = a/a_m_/c_e_.(24)

Recalling that, at low solution concentrations, the Langmuir equation is a = a_m_ K_L_ c_e_ and (after a rearrangement) K_L_ = a/a_m_/c_e_, one can define K_0_ directly by K_L_. In such approach, K_0_ does not depend on an extensive value of adsorption capacity. Basically, the ΔH° value calculated from the slope of the temperature dependence of log(K_0_) should be identical regardless of the definition of K_0_, provided a_m_ is temperature invariant. As in most of the adsorption systems, a_m_ varies with adsorption temperature, ΔH° calculated from K_D_ is higher than that calculated using K_L_ by the slope of log(a_m_) vs. 1/T dependence. For the above reasons, we prefer using K_0_ = K_L_. It is worth noting that adsorption constants calculated from various adsorption equations have been also applied for estimation of thermodynamic parameters [76].

Values of free enthalpies for the studied adsorbents calculated using different modes of the K_0_ estimation described above, along with correlation coefficients for linear fits of experimental points to Equation (17) are shown in Table 8.

The values of free enthalpies calculated for a given adsorbent markedly differ depending on the mode of K_0_ estimation, which shows dangers and uncertainties that may occur in reports on thermodynamic properties of adsorbing systems. We raise the following question, which values are the most reliable?

It has been commonly accepted that adsorption of anions decreases with increasing temperature and as a result anion adsorption is exothermic, while metal cation adsorption increases, and the adsorption is endothermic [77,78,79]. From our data, one may conclude that arsenate anion adsorption is endothermic. However, endothermic adsorption of anions has also been frequently reported: for phosphates on modified chitosan beads [80], for sulphate on magnetite, activated carbon and their composites [81], for arsenates on iron modified bentonite [82]. The increase in temperature increased nitrate and phosphate adsorption on chitin, plant particles and different kinds of anion exchange resins [82,83,84,85,86,87]. Xi et al. [88] observed that the adsorption of Sb(III) anions on bentonite was exothermic, while the adsorption of Sb(V) anions on the same bentonite was endothermic. The adsorption enthalpies are often ±20 kJ/mole or greater [77]; however, significantly lower values were noted as well [78,79,80,81,82,83,84,85,86,87,88]. The positive values of the free enthalpy of the adsorption process have to be connected with an increase in the entropy of the system. The latter effect may be explained by the adsorbed As(V) anions replacing the other anions previously neutralizing the adsorbing surface. Additionally, water molecules present at the surface and hydrating As(V) anions may be redistributed.

Table 6 shows the values of equilibrium As(V) adsorption in kinetic measurements measured after 5 h of equilibration. However, these values ought to be established after an infinite time at the respective equilibrium concentration, c_e_. This c_e_ value can be found knowing the initial concentration applied for kinetic measurements, c_0_, the amount of the adsorbate adsorbed at the kinetic equilibrium, q_e_, the volume of the solution, V, and the mass of the adsorbent, m:c_e_ = c_0_ − m q_e_ V^−1^.(25)

The value of q_e_ for the given equilibrium concentration should correspond to the value of adsorption a determined from the adsorption isotherm at the same concentration (and at the same temperature of the measurements). To compare the above values, we calculated c_e_ for each adsorbent at each of the three initial adsorbate concentrations and estimated the corresponding adsorption values, a, from the respective isotherms. Two modes of the a values estimation were applied:The a values were linearly interpolated from the respective isotherms at a given c_e_.The a values for a given c_e_ were calculated from the Langmuir adsorption equation using adsorption capacity (am) and K from Table 5.

Comparison of the values of adsorption established from kinetics and from equilibrium measurements at the same adsorbate concentrations is shown in Figure 12. The linear regression lines were forced through zero.

The equilibrium adsorption values derived from kinetic studies are much closer to the values of equilibrium adsorption directly approximated from adsorption isotherms than to the adsorption values derived from the Langmuir equation. This indicates that the latter mode of data acquisition is more sensitive to approximation errors. Indeed, the RRMSE values (Figure 8) are very high at low adsorbate concentrations—corresponding to these at which the kinetic studies were performed. One can suspect that the equilibrium adsorption values estimated from kinetic curves should be more similar to those derived from adsorption isotherms. A better coincidence was reported by Ali et al. [89] for the biosorption of heavy metals by non-living algae cells.

The adsorption rate constants calculated at various initial solute concentrations were different, similarly as it has been reported by plenty of scientific reports. We thought that the scaling of the LHS of the classical PSO kinetic equation by q_e_:dq_t_/(q_e_ dt) = k(q_e_ − q_t_)^2^(26)
would lead to calculation of similar rate constants. They were closer indeed, but not acceptably similar. Note that Equation (26) is formulated basing on the reaction rate expressed according to the physicochemically correct definition of the reaction rate as the change in the extent of the reaction, α, in time dα/dt where α = q_t_/q_e_. Calculation of different rate constants at different concentrations was considered by Huang et al. [90] as an artifact caused by erroneous mode of data regression with the linearized kinetic model. Accounting for the concentration dependence of q_e_ in the model equation, they proposed a rectified approach in which the same relative values of kinetic constants were obtained independently on the initial solute concentration.

### 4.3. Relations between Physicochemical Parameters of the Sorbents and Anion Adsorption Characteristics

Table 4 and Table 5 show surface charge at pH = 6 estimated from titration curves and As(V) anions adsorption capacity measured at the same pH, respectively. Since surface charge governs the ions’ electrostatic adsorption, these values should be more or less equal, contrary to what is shown in Figure 13.

For temperature activated fly ashes, the charge of adsorbed As(V) is higher than that measured by titration. Here, we recall that because the initial pH of the titration (pH = 5) is lower than the pH of the charge measurement (pH = 6), some of the finest particles of Si and/or Al oxides, which have the highest charge, may still dissolve and the measured adsorbent charge may be lower. The excessive adsorption of As(V) anions may be also due to their very strong affinity to fly ash surfaces by forming surface compounds. In such case, the surface should recharge (change the charge sign), which could be measured by zeta potential—it is worth checking. In contrast, for all chitosan-containing adsorbents, the charge of the adsorbed As(V) anions is lower than that measured by titration. Most probably, the titration overestimates the real surface charge, as mentioned before. Furthermore, chitosan modification may decrease As(V) adsorption on fly ashes by covering a part of their adsorbing surface.

Figure 14 shows linear correlation coefficients between all parameters of the studied adsorbents along with the direction of the respective dependencies expressed by signs of their slopes.

The upper triangle drawn with thick lines shows correlation coefficients between surface parameters and the lower, smaller triangle shows correlation coefficients between adsorption parameters. The most relevant factor to the main story are the correlations between surface and adsorption parameters, which are presented within a rectangle located between the above triangles. We expected that the surface area (S) of the adsorbents would correlate with surface charge (Q) and with As(V) adsorption capacity (a_m_); however, no correlation between these values was found. For all adsorbents, both the As(V) adsorption capacity and surface charge (especially the charge measured after the adsorbents’ pretreatment at pH = 5, Q_pH6_(5)) correlate well with mesopore volume measured from MIP. We cannot find any physicochemical principle responsible for the above dependence. Similar fake dependencies occur for the free enthalpy of As adsorption and mesopore and material pore radii. Higher adsorption constants, K, occur for adsorbents with more flat surfaces (smaller fractal dimensions, D, calculated from adsorption isotherms), which appears to be rationale.

## 5. General Comment

Recently, the adsorption of arsenic using natural materials or waste products from industrial or agricultural operations has emerged as an option for developing economic and eco-friendly wastewater treatment processes. Numerous low-cost adsorbents have so far been studied for the removal of arsenic from water and wastewater red mud, natural minerals, blast furnace slags, hydrocalcites, hydroxides and various bioadsorbents: dry plants, fish scale, orange juice residue, tea fungal biomass, rice polish, bone char, leather waste, calcium alginate and many others, for which the maximum As(V) adsorption was in the range 0.022–132 mg/g (at variety of pH and initial concentrations) [91,92,93]. Among a lot of adsorbents reviewed in the above papers, chitosan is mentioned twice (adsorption of As(V) equal to 14 and 58 mg/g for initial concentrations 10 and 400 mg/L, respectively) whereas fly ash was not mentioned. Diamadopoulos [94] measured As(V) adsorption on lignite-based fly ash ranging from 8 to 30 mg/g with decreasing pH of the medium from 10 to 4. Vivek and Nalini [95] reported arsenic adsorption from 2 to 22 mg g^−1^ onto Fe-chitosan composites; however, Gupta et al. [96] reported extremely high arsenic adsorption (119 mg/g) onto Fe-Chitosan nanospheres. To achieve an easier comparison with the above data, the adsorption capacities for the sorbents studied in this paper are: fly ash ~50 mg/g; chitosan ~30 mg/g and fly ash modified with chitosan ~25 mg/g. Generally, the As(V) adsorption on a large number of waste derived sorbents (including the studied sorbents) is only a few times lower than on the best synthetic resins (289 mg/g as reported by Qureshi et al. [97]).

## 6. Conclusions

Highly expensive and energy consuming thermal treatment of fly ash does not improve its physicochemical properties. It appears not to be a “thermal activation”, but rather deactivation.

Despite the apparently high surface charge of chitosan, chitosan–fly ash composites do not adsorb arsenate anions better than fly ash itself. Due to higher porosity of their beds, the composites may be more effective in column adsorption processes.

Anion adsorption parameters were practically not related to physicochemical properties of fly ash and chitosan derived adsorbents.

The moderate adsorption capacity along with the high availability and low price of fly ash and chitosan derived adsorbents make them promising for arsenic removal from contaminated waters.

Estimation of free enthalpies of adsorption basing on Langmuir adsorption constant was proposed.

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
