# Peer review of "Arsenate Adsorption on Fly Ash, Chitosan and Their Composites and Its Relations with Surface, Charge and Pore Properties of the Sorbents"

_materials, 2020, doi:10.3390/ma13235381_

Round 1
Reviewer 1 Report
This manuscript reports the investigation of As(V) adsorption from aqueous solutions using waste-derived sorbents, i.e., fly ash and chitosan. The study is interesting and makes contributions to the field. I would suggest major revision before considering publication on the journal Materials. The followings show my major concerns:
- The title is too long and needs to be shortened and concise.
- Line 72, the experimental detail about the coating process should be included here.
- For sections 2.4.1 and 2.4.2, those two characterization tests are quite common and their mechanisms are common knowledge, so it is not suitable to have that much explanation there in a research paper.
- where is section 2.1???
- If possible, please use the professional expression for all equations.
- line 191, since ref 63 is cited here, it would be better to add a short explanation why the K in Equation (15) can be represented by the K in Equation (14).
- Although the pseudo-second-order kinetic model is widely used, misapplications of those equations have been found (AIChE Journal, 64(5) 2018, 1793, DOI 10.1002/aic.16051). I suggest the authors add a short discussion about this issue by citing the highlighted paper. As a matter of fact, the authors have discussed this issue from lines 450 to line 473.
- "SEM images" is not a proper section title. Please revise it accordingly.
- The scale bar and portions of the text below in the SEM images (FA and FACS) are missing.
- Line 336, "To calculate the..." this sentence is not clear. Please clearly explain how the authors treat and calculate the data.
- For Fig 7 and 9, please add proper text in both x and y-axis. In addition, error bars for the data in these two figs should be added.
- In fig9, please fix the format issue of the legend.
- The present investigated the possibility of applying the waste-derived fly ash for As adsorption. It's of importance regarding the resuing the waste and sustainability. However, it seems that the adsorption performance is not quite promising. It would be better if the comparison with other waste-derived sorbents reported in the literature is shown and listed here.
Reviewer 2 Report
The manuscript by Adamczuk and co-workers details the preparation and adsorption application, as well as characterization of waste-derived sorbents. The topic is relevant to both academics and industrial professionals, and it is of interest to the readers of Materials, and fits well the scope of the journal. Overall the manuscript is fairly well-written, however there are several minor and major issues that should be addressed prior to further consideration by the journal.
1, The authors imply that chitosan is a green material but its derivation from chitin requires harsh conditions including both HCl and NaOH treatment. The drawbacks of chitosan should also be mentioned.
2, Avoid grouping large number of references. Mention the works individually if they are important enough or omit them.
3, The authors should make suggestions about increasing the competitiveness of the materials as they adsorption capacity is low.
4, The data reported in the manuscript have no errors reported. The authors should provide errors to demonstrate the reproducibility of the results (using independently prepared materials), or at least discuss reproducibility issues and key points for the work to be easily taken up by the community.
5, The scale bars on the SEM images are not visible. Add a decent sized scale bar for all images.
6, How was the degree of deacetylation of chitosan obtained? What are these two values 70,92%? Should be 70.92%? The determination of deacetylation is notoriously difficult and the accuracy is usually within 5-10%. However, the authors reported the value down to 2 decimal places which seems erroneous. Reveal the technique and the accuracy.
7, The introduction is very short, it reads as a chemical abstract with too many references and no context or proper background to the topic. It is also too specific without placing the research into context.
8, For instance utilization of waste should be discussed in a paragraph and broad examples given to demonstrate the importance of this expanding field (DOIs 10.1021/acssuschemeng.9b04245, 10.1021/acssuschemeng.0c05906, 10.1039/D0GC02580A).
9, It is necessary to add an initial table revealing the difference between FA, FA500, FA700, FA900, CS, FACS, FA500CS, FA700CS and FA900CS. This would help the reader to follow the manuscript and would provide a quick reference to the materials.
10, Compared to the length of the results and discussions, the conclusion section is too short and vague. The main research findings should be summarized in quantitative statements. Future directions should be given, as well as the drawbacks and limitations of the methods discussed.
11, The title needs to be more concise as it is currently too long and specific.
Reviewer 3 Report
Review for materials-1000030
I have attached a scanned version of the manuscript with annotations and some numbered items #i. To these I will refer in the following.
#1 already at this point the outcome of that modelling is no surprise. Langmuir often fits and more surprisingly to me, all kinetic data modelling I come across fits to the pseudo second order model. Maybe it is time to think about why this is the case. In general, since the authors have titrated their solid, this approach is too simple. At least for RT they could have developed a surface complexation model.
#2 what kind of properties? Just surface area and pore diameters?
#3 Titanium will be far too expensive for these puposes.
#4 why are you using sulfate? Sulfate quite strongly adsorbs to some oxides, so it might not only fix the ionic strength here, but at the same time adsorb and be thus a (weak) competitor. I remember there is a paper on phosphate adsorption to goethite, where it was shown that sulfate may compete. Phosphate is very similar to arsenate.
#5 what does the toothpaste mean?
#6 do not understand the context here.
#7 the authors need to explain how they calibrated their pH measurement set-up, then the absolute amount of solid suggests that in some cases less than 10 square meters were inserted. Under such conditions it is difficult to get good data with high titrant concentrations (100 mM here). Furthermore, I think the conditions are ill-chosen, since in the titrant there is such an excess of sodium. Even from that point of view 10 mM NaOH would have been much better.
#8 was this a particular fly ash and chitosan sample or do the cited values hold for any sample? I would not believe there is a generally applicable value here.
#9 it is a Langmuir constant not more. I find it very difficult to relate it to thermodynamic quantities since the electrostatic contribution that varies with conditions (and might vary here with arsenate concentration) affects everything and is not accounted for, which would be possible with a more comprehensive model (see #1).
#10 Please explain sample ID even here. Also scales should be made clearer.
#11 these charges should be per surface area. Then they can be compared correctly amongst each other and with other sorbents.
#12 I suspect the lower values at pH 3 are a consequence of the ill chosen conditions in the titrations. I have seen such effects even with very sophisticated titrators. For example at low pH the liquid junction effects can interfere in those. Then depending on how well you calibrate everything you can get a drop, a constancy or an increase in proton uptake.
#13 The trend is opposite to some anion data and to expectations. The trend is not discussed in that sense. For cation adsorption most people expect increase with temperature. For anions the expectation is inverse, but some data show the trend observed here. The pH is not specified in the caption. All conditions should be given. X-axes are somewhat difficult. They start at zero, zero for all cases I guess?
#14 how do these values compare to anion adsorption observed for other systems, also with surface complexation models? There should be something in the literature.
#15 here I wondered how the values from Langmuir and kinetics compare, but this is shown later. More discussion at this point is warranted.
Langmuir parameters and most other model parameters should be on surface area basis (like the maximum uptake etc.) and molar units. The comparisons would potentially change conclusions, but these scales are the relevant ones, since we deal with surface adsorption! So prior to a revision along those lines, I would not recommend publication. I like some aspects of the work, like the final figure 12. I would hope the work would be improved by the above comments.

Reviewer 4 Report
The title is clear. The content is in accord with title.
The manuscript adheres to the journal's standards.
The size of the article is appropriate to the contents.
Because there are a lot of researches in this topic, the authors must underline the major findings of their work and explain how the use of their proposed procedures and activated fly ash coated with chitosan materials represents a progress to other similar published papers. Comparison with other publication is necessary.
The Abstract is OK.
The key words permit found article in the current registers or indexes.
In the introduction it is clearly described the state of the art of the investigated problem.
The methods are well described and the equipment and materials have been adequately described.
The paper was written in standard, grammatically correct English, small corrections are necessary.
Please use the u.m in S.I.
Please verify notation for CSF (Table 3… not subscript).
Table 5 and Table 7 must be rearranged … is relatively hard to understand.
The figures have a good quality. Please provide high quality for Figure 12.
The tables contain necessary results.
The Conclusion must be completed with the major results, it is not been justified sufficiently.
Please provide minimum 2 references from this journal (last years), for demonstrated that manuscript is in Materials topic.
The literature is sufficiently critical, current, and internationally evaluated
The paper has the text presented and arranged clearly and concisely.
Please verify the reference; there are small mistakes (21, 36, 37).
Round 2
Reviewer 1 Report
The current version is ready to publish.
Reviewer 2 Report
The authors have done a thorough revision. However, there are multiple typos and inconsistencies in editing, and therefore the manuscript needs to go through a profound proofreading.